# Exploitation of the Timing Capabilities of Metallic Magnetic Calorimeters for a Coincidence Measurement Scheme

Philip Pfäfflein [1,2,3,*], Günter Weber [1,2], Steffen Allgeier [4], Sonja Bernitt [1,2], Andreas Fleischmann [4], Marvin Friedrich [4], Christoph Hahn [1,2], Daniel Hengstler [4], Marc Oliver Herdrich [1,3], Anton Kalinin [2], Felix Martin Kröger [1,3], Patricia Kuntz [4], Michael Lestinsky [2], Bastian Löher [2], Esther Babette Menz [1,2,3], Uwe Spillmann [2], Binghui Zhu [1,3], Christian Enss [4,5] and Thomas Stöhlker [1,2,3]

1   Helmholtz Institute Jena, Fröbelstieg 3, 07743 Jena, Germany
2   GSI Helmholtzzentrum für Schwerionenforschung, Planckstraße 1, 64291 Darmstadt, Germany
3   Institute for Optics and Quantum Electronics, Friedrich Schiller University, Max-Wien-Platz 1, 07743 Jena, Germany
4   Kirchhoff Institute for Physics, Heidelberg University, Im Neuenheimer Feld 227, 69120 Heidelberg, Germany
5   Institute for Data Processing and Electronics, Karlsruhe Institute of Technology, Hermann-von-Helmholtz-Platz 1, 76344 Eggenstein-Leopoldshafen, Germany
*   Correspondence: p.pfaefflein@hi-jena.gsi.de

**Abstract:** In this report, we compare two filter algorithms for extracting timing information using novel metallic magnetic calorimeter detectors, applied to the precision X-ray spectroscopy of highly charged ions in a storage ring. Accurate timing information is crucial when exploiting coincidence conditions for background suppression to obtain clean spectra. For X-rays emitted by charge-changing interactions between ions and a target, this is a well-established technique when relying on conventional semiconductor detectors that offer a good temporal resolution. However, until recently, such a coincidence scheme had never been realized with metallic magnetic calorimeters, which typically feature much longer signal rise times. In this report, we present optimized timing filter algorithms for this type of detector. Their application to experimental data recently obtained at the electron cooler of CRYRING@ESR at GSI, Darmstadt is discussed.

**Keywords:** timing; coincidence; metallic magnetic calorimeter; microcalorimeter; precision X-ray spectroscopy

## 1. Introduction

X-ray spectroscopy of characteristic transitions is an indispensable tool for the investigation of atomic structure in heavy ionic systems. In these systems, inner-shell electrons experience electromagnetic field strengths many orders of magnitude higher than what is achievable by the most powerful laser systems and magnets. For the probing of quantum-electrodynamics as well as relativistic effects, $K_\alpha$ ($n = 2 \rightarrow n = 1$) transitions are of particular importance. In systems with the highest atomic numbers ($Z$), these transitions between the most strongly bound states reach energies up to about 100 keV [1]. For studies of the heaviest, few-electron ions, storage rings are particularly well suited. Two examples are Experimental Storage Ring (ESR) [2] and CRYRING@ESR [3], both located at GSI, Darmstadt. In these devices, the application of stochastic cooling and/or electron cooling decreases the emittance as well as the kinetic energy dispersion [4] of the ion beam. This leads to a well-defined Doppler shift as well as a reduced Doppler broadening of the spectral lines emitted by the projectiles. Moreover, collision experiments benefit from the use of in-ring targets exploiting the high repetition rates in the order of 1 MHz, resulting in a superior luminosity compared with single-pass setups. This enables the use of dilute gas targets [5], providing single-collision conditions, thus yielding X-ray spectra undistorted by multiple-collision effects.

Precision X-ray spectroscopy further benefits from recent advances in the development of so-called microcalorimeter detectors. These are based on arrays of low-temperature detectors (LTDs) for ionizing radiation, which measure the temperature increase in a small absorber volume upon absorption of an incident X-ray. In the photon energy regime of few to tens of keV, various LTD technologies have demonstrated resolving powers significantly better than $1 \times 10^3$ [6–11]. We refer to the resolving power as $R = E/\Delta E$, where $E$ is the measured energy and $\Delta E$ is the full width at half maximum. This is an improvement of the spectral resolution by more than one order of magnitude compared with commonly used semiconductor detectors based on silicon or germanium. At the same time, microcalorimeter detectors retain the broad bandwidth acceptance of solid-state detectors. This is in sharp contrast to the narrow spectral acceptance typical of crystal spectrometers. The unique combination of high spectral resolution and acceptable quantum efficiency over an extended range of photon energies makes the LTD a particularly promising type of detector system for the scientific program of the SPARC collaboration [12]. Prototypes of such detector systems have been developed and were recently deployed in several test experiments at GSI [13–15].

However, beside a sufficient spectral resolution, a good signal-to-background ratio, regularly referred to as signal-to-noise ratio (SNR), is also key to most high-precision X-ray spectroscopy measurements. In experimental settings where the process under investigation is not already dominant in the considered spectral region, the SNR can often significantly be improved via background suppression. An obvious route to discriminate the photons of interest from unrelated radiative processes is the so-called coincidence technique. It relies on the time-resolved detection of various reaction products to apply temporal constraints. Recently, we succeeded in the implementation of such a time-resolved measurement using novel microcalorimeter detectors at CRYRING@ESR [16]. In the following, we discuss the details of extracting timing information using the new type of detector. This was performed in the post-processing of the recorded pulses. We compare the performance of two different algorithms for that purpose.

## 2. The Experiment

The data presented in this work was obtained at the electron cooler of CRYRING@ESR. Two microcalorimeters for precision X-ray spectroscopy were positioned at 0° and 180° with respect to the ion beam axis. In the cooler section, stored $U^{91+}$ ions with a kinetic energy of 10.225 MeV/u interacted with the cooler electrons. In these collisions, a free electron can recombine with an ion under the emission of a photon via a process referred to as radiative recombination (RR) [17]. Electrons with a relative velocity close to zero with respect to the circulating ions tend to recombine into Rydberg states, i.e., those with high quantum numbers $n, l$, and subsequently decay to the ground state via radiative cascades [18,19]. As a consequence, the formation of a $U^{90+}$ ion in the cooler section yields two types of particles of interest: a down-charged $U^{90+}$ ion and one or more photons. The ion was detected by an ion counter downstream behind a dipole magnet. The magnet separated the charge-changed ions from the primary $U^{91+}$ beam. The photons in the X-ray regime were recorded using microcalorimeter detectors of the maXs type (Microcalorimeter Arrays for High Resolution X-ray Spectroscopy) [13,20]. The used maXs-100 absorber arrays were tailored to deliver a spectral resolution $\Delta E_{FWHM} < 50$ eV for photons between a few keV and more than 100 keV energy. For a detailed description of the experimental setup, the reader is referred to [16].

For photons and down-charged ions resulting from the same RR reaction, the difference in their arrival times is determined by the distance of the interaction from the respective detector and the ion beam velocity. The possible range of these so-called time-of-flight (TOF) values is limited by the lifetime of the relevant excited states in $U^{90+}$ and the flight time through the cooler region. For the present experimental conditions the width of this coincidence window is below 100 ns (see [19] for details). By setting a corresponding condition on the observed arrival time difference, one can achieve an effective suppression

of background radiation in the photon spectra. This requires sufficient time resolution of the detector systems. This is easily fulfilled by the ion counter. It is based on the detection of secondary electrons produced from ion impact on a baffle plate using a channeltron with a very fast signal rise time. Likewise, semiconductor detectors for X-ray detection routinely achieve time resolutions down to a few tens of nanoseconds. Thus, coincidence measurement between X-rays and ions is a well-established experimental technique. However, the microcalorimeters employed in the present study featured a much longer rise time in the order of 10 μs. Recently, coincidence measurements have been realized using MMCs [16] as well as transition edge sensors [11].

In the present experiment, the voltage signals of the 32 channels of the MMC detectors were digitized using Struck SIS3316 modules. Once the digitizers registered a pulse, the readout of the data acquisition system was triggered. A signal trace of about 2 ms in length, consisting of $2^{14}$ voltage values, was stored as an event within a file. For each of these photon hits, the digitized signal of the particle detector trigger was also written into the same readout event within a 0.5 ms window centred around the time of the photon trigger. For a detailed description of the detector readout and data acquisition, the reader is referred to [16]. Storing the raw detector pulses allows a software-based coincidence scheme to be implemented during the post-processing of the data. Due to the long rise-time of the MMC pulses and a wide range of different pulse amplitudes, a dedicated timing filter algorithm is necessary to extract information on the photon arrival time with a reasonable temporal resolution. The performance of two timing filter algorithms in their application to background suppression via a coincidence condition is compared in the following section.

## 3. Coincidence-Based Background Suppression

With trigger, we refer to any logic that is capable of identifying an event, such as a pulse in a signal trace, and returning its timing information. For the present detector pulses, two different trigger algorithms were implemented. The first was a leading edge discriminator (LED) with an adaptive threshold, also referred to as a $k\sigma$ filter. The second was a trigger filter reproducing the functionality of a constant fraction discriminator (CFD). The threshold of the LED was set to a fixed factor $k$ times standard deviation $\sigma$ as the square root of the variance of the signal trace prior to the current sample. In previous microcalorimeter measurements, which did not require precise timing information, the $k\sigma$ trigger was used to determine the time of arrival of the photon with better accuracy than the hardware trigger. The application of the $k\sigma$ logic to the band-pass filtered signal can swiftly and reliably identify both isolated hits and multiple consecutive hits. The latter result in—compared to single hit pulses—distorted pulse shapes. The band-pass filter is implemented as a box filter. For a discrete signal $s_j$, the $i$-th value of the box filter of width $w_B$ is defined as $box_i = \sum_{k=i-2w_B+1}^{i-w_B} s_k - \sum_{k=i-w_B+1}^{i} s_k$ (see [21] for details). The timing information returned by the $k\sigma$ trigger, however, is insufficient to achieve acceptable background suppression in coincidence measurements. The CFD filter was, therefore, introduced as a robust alternative for the extraction of timing information. It is based on the idea that the sum of the signal and an inverted, scaled, and delayed copy of the signal have a zero crossing at a fixed fraction of the pulse height. This is true for signals with the same rise time but a severely different pulse height. The implementation and application of both algorithms is discussed in more detail in [22].

In Figure 1, we present a typical detector pulse together with the results of the application of both trigger filters. There, the dashed vertical line indicates the determined arrival time. As $\sigma$ is governed by the noise floor and $k$ is held constant, in the case of the $k\sigma$ filter, this time depends on the signal height, as pulses with high amplitudes cross the set threshold earlier than smaller pulses. In contrast, the CFD filter pinpoints the arrival time when the pulse reaches a fixed fraction of its height. The identified arrival time is, therefore, ideally independent of the absolute pulse amplitude.

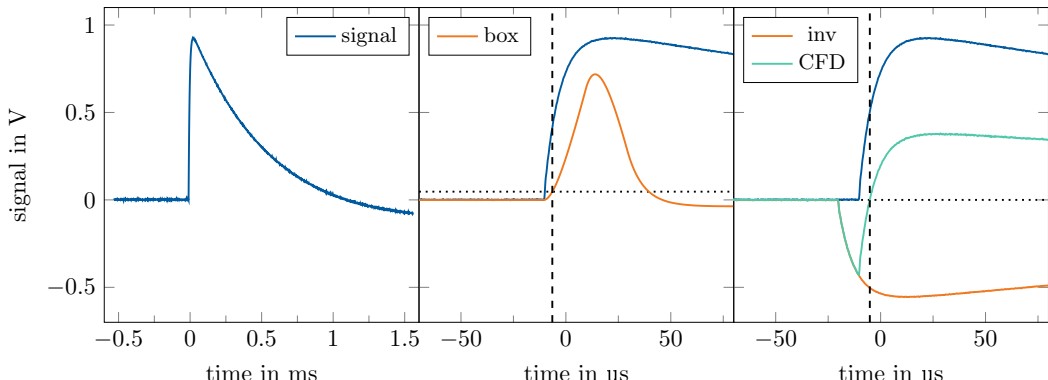

**Figure 1.** Illustration of the $k\sigma$ and CFD filter algorithms. Horizontal dashed lines represent threshold levels that the filtered signal is required to cross in order to initiate a trigger. Vertical dashed lines indicate the resulting arrival time of the pulse. Left: Entire raw detector signal as recorded by the data acquisition system ($t = 0$ defined by the hardware trigger). Blue line in all plots: raw signal. The middle and right plots focus on the signal's rising slope. Centre: Illustration of the $k\sigma$ trigger applied to the box-filtered signal (orange; see text for details). The threshold is set as a $k$-multiple of standard deviation $\sigma$ of the filtered signal trace ($k$ is drastically exaggerated for illustrative purposes). Right: The CFD signal (green) is the sum of the raw signal (blue) and a scaled, delayed, and inverted copy of the raw signal (orange).

The different behaviour of both timing filter algorithms is contrasted in Figure 2. Here, the TOF between X-ray photons recorded by the maXs detector located at the 180° port and down-charged ions is presented for four transitions in $U^{90+}$ with energy values between 14 and 87 keV in the laboratory system. It is obvious that the TOF resulting from the application of the $k\sigma$ trigger yielded a broader distribution, i.e., worse timing resolution, than the CFD approach, an effect that was particularly pronounced for low energy values. In addition, the four photon energy values corresponded to different pulse amplitudes; consequently, the $k\sigma$ filter yielded a different position of the coincidence peak for each energy value. In contrast, the application of an additional CFD trigger led to constant timing throughout the entire spectrum. The slightly deteriorating time resolution at lower photon energy values can be explained by the larger influence of electronic noise at small pulse amplitudes.

In order to benchmark the filters with respect to the aforementioned coincidence technique for background suppression, we applied a time condition corresponding to the width of the coincidence peaks in Figure 2. The resulting X-ray spectra are presented in Figure 3 for the detector located at 180° at the CRYRING@ESR electron cooler. The energy region displayed contains the L→K transitions as well as the K-RR radiation (direct recombination into the K shell). As discussed above, the beneficial characteristics of the CFD filter allowed a narrower coincidence condition to be achieved, yielding a further reduction in the background of the recorded spectrum. In fact, the CFD approach achieved an improvement by more than a factor of five over the $k\sigma$ filter. According to the present data, the CFD filter enabled almost background-free spectra when used in combination with the particle detector signal for the determination of the time of flight and the subsequent application of a coincidence condition.

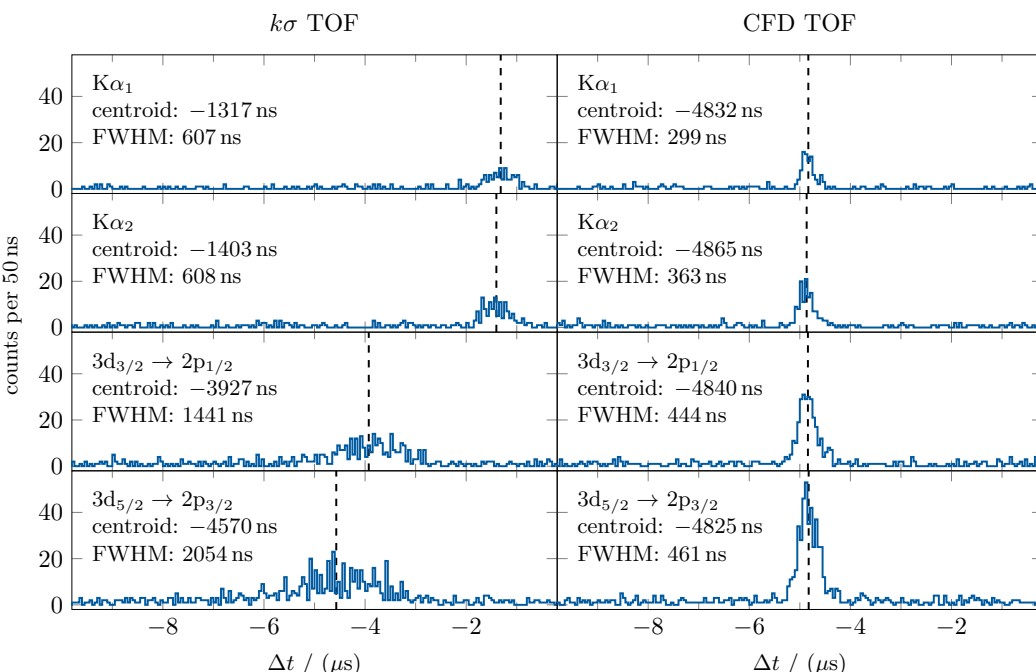

**Figure 2.** Time-of-flight (TOF) spectra of photons emitted from $U^{90+}$ (in four distinct regions of interest) relative to the arrival time of the corresponding down-charged ions. Data recorded by the 180° microcalorimeter and the particle detector. The data resulted from the application of the $k\sigma$ (left) and the CFD trigger (right). Noteworthy differences are the broader range of arrival times of the $k\sigma$ trigger, i.e., worse timing resolution, than that of the CFD trigger, as well as a pulse-amplitude-dependent shift in the mean arrival time recorded by the $k\sigma$ trigger. See text for details.

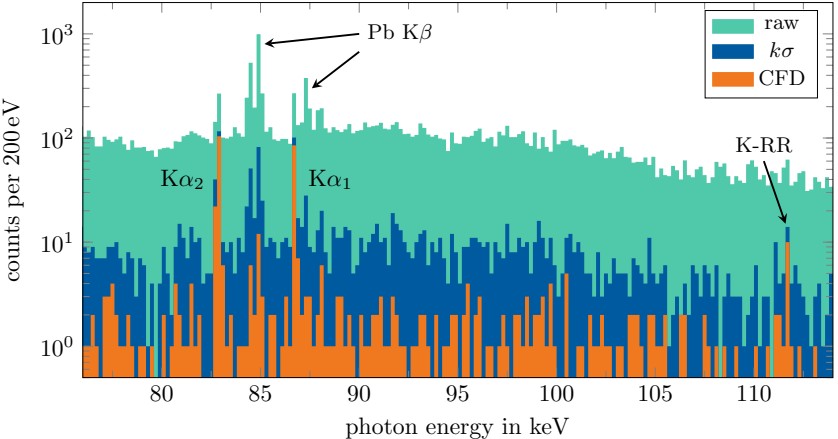

**Figure 3.** Demonstration of the background reduction achieved by means of a coincidence measurement. The spectrum without application of a coincidence condition is marked as 'raw' and compared with the spectra with an appropriate coincidence condition for the $k\sigma$ trigger and the CFD trigger, respectively. The distinct spectral features are labelled. Those stemming from $U^{90+}$ appear prominently in the coincident spectra.

## 4. Conclusions

The performance of two timing filter algorithms applied to coincidence measurements using novel microcalorimeter detectors was demonstrated. A filter mimicking the functionality of a constant fraction discriminator was found to yield time resolution of a few 100 ns, allowing a background suppression improvement by a factor of five to be obtained when compared with the previously used trigger filter. This was sufficient to mask almost all background radiation in the example case of a recent experiment at the electron cooler of

CRYRING@ESR. Currently, we are investigating whether the time resolution can be even further improved using a filter that contains information of the pulse shape.

**Author Contributions:** Conceptualization, T.S., G.W., M.O.H. and P.P.; software, M.O.H. and P.P.; formal analysis, P.P. and M.O.H.; investigation, all; resources, C.E. and T.S.; writing—original draft preparation, P.P. and G.W.; writing—review and editing, all; visualization, P.P.; supervision, T.S.; project administration, G.W.; funding acquisition, C.E. and T.S. All authors have read and agreed to the published version of the manuscript.

**Funding:** This research was funded by European Research Council (ERC) under the European Union's Horizon 2020 research program as well as by the innovation program (grant No. 824 109 "EMP"). B. Zhu acknowledges CSC Doctoral Fellowship 2018.9—2022.2 (grant No. 201 806 180 051). We also acknowledge the support provided by ErUM FSP T05—"Aufbau von APPA bei FAIR" (BMBF No. 05P19SJFAA and No. 05P19VHFA1).

**Data Availability Statement:** The data presented in this study are available upon request from the corresponding author. The data are not publicly available due to ongoing analysis.

**Acknowledgments:** This research was conducted in the framework of the SPARC collaboration, experiment E138 of FAIR Phase-0 supported by GSI.

**Conflicts of Interest:** The authors declare no conflict of interest.

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
