# Peer review of "Exploitation of the Timing Capabilities of Metallic Magnetic Calorimeters for a Coincidence Measurement Scheme"

_atoms, doi:10.3390/atoms11010005_

Round 1

Reviewer 1 Report

The authors describe a method for improving microcalorimeter SNR by using photon-ion coincident measurements to obtain pure x-ray spectra and compare the differences between leading timing and constant fraction timing. Despite the well-known fact that constant fraction timing methods outperform leading timing in nuclear and atomic physics experiments, considering that this is a relatively new (perhaps not the most recent) report on coincident measurement with microcalorimeters, I recommend that it be published in ATOMS. In addition, I suggest the authors clarify the following technical details in the manuscript.

1.     Is the overall trigger signal (data recording) produced by the microcalorimeter or by the ion detector, or is it the result of an AND of the two? Is it direct hardware coincident, or do photon and ion events coincident correlate by timestamps? Because the ion detector signal is much faster than the microcalorimeter signal, consider that despite the ions arriving at the detector after the photons, the output of the electronics may precede the x-ray photons, and the x-ray signals are far more numerous than the ions. How are the authors going to bridge the gap between the two? In summary, given this is an article describing a technique for coincident signals, it is suggested that the authors provide sufficient technical details, such as a logic flowchart of the coincidence, including the delay, broad and manipulation of each signal, as well as the model number of the electronics components used.

2.     Is the signal in Figure 1 obtained using an oscilloscope, and why is the time start point of the signal less than zero?

3.     Why is the CFD signal taken as a scaling inversion of the original signal and then added to it rather than passing the original signal directly to the discriminator and then outputting the logic signal? Because both scaling and inversion cause delays in the signal and introduce additional errors, the TOF may be broadened further, deteriorating the time resolution.

Author Response

Dear reviewer,

thank you very much for the kind comments. We addressed them as follows:

  1. The data recording is triggered by a photon event. We introduced more details of this process in line 93 ff. As all of the triggers are implemented in software, we did not include a flow chart. If it is considered necessary beyond the explanation, please tell us.
  2.  It is one of the raw signals as recorded by the data acquisition system. We included more details in the caption of figure 1.
  3.  The manipulation is done in post processing in our analysis software routine. Scaling, delay and inversion were mentioned to introduce the basic idea of the CFD.

Kind regards

Reviewer 2 Report

This paper reports on a comparative study of two filter algorithms to extract timing information from output signals of metallic magnetic calorimeters (MMC). For precision x-ray spectroscopy of highly-charged ions, coincidence measurement is an important technique to improve the signal-to-noise ratio (SNR), and thus accurate timing information plays a crucial role. The authors developed optimized timing filter algorithms based on the functionality of a constant fraction discriminator (CFD) and succeeded in improving time resolution of MMCs by more than a factor of five, compared with the standard kσ filter algorithms. The authors measured the radiative recombination (RR) x-ray spectrum from U90+ by applying the coincidence condition with the CFD timing filter, which showed the CFD filter can achieve almost background-free x-ray spectra.

Improvement of time resolution of microcalorimeter detectors is one of the remaining problems to be solved for application to atomic physics. Because the result reported in this work is the important progress toward this goal, I recommend this paper be accepted to the Atom journal. 

The followings are minor points to be revised: 

  • Line 35, “1×103”: the definition of resolving power is required.
  • Information on the sampling rate of the maXs-100 system is helpful for readers though it is written in Ref [15]. 
  • Line 89-90, “until the recent experiment [15] a coincidence scheme has never been realized using such detector systems.”: For other types of microcalorimeters, such as superconducting transition-edge sensors (TES) microcalorimeters, time information was employed to improve the SNR for precision x-ray spectroscopy, e.g., see Hashimoto et al. Phys. Rev. Lett. 128, 112503 (2022). I recommend that, here, authors should specify MMCs, rather than general microcalorimeters.
  • Fig.1: the meanings of the orange and green curves are not well described. Authors should add more explanations about the box filter and the CFD filter in the caption or the main text.
  • Line 103: definitions of σ and k are necessary (explanations in line 116 are not enough.)
  • What is the limit of time resolution which can be achieved in this system? 

Author Response

Dear reviewer,

thank you very much for the kind comments. We addressed them as follows:

  1. The definition of the resolving power has been introduced in line 36. The length in samples and time has been added in line 95f.
  2. Line 91f has been changed to cite the success of coincidences with the different detector techniques.
  3. The caption of figure 1 was changed to include more detail and the definition of the box filter has been included in the text: line 119.
  4. The meaning of k and sigma have been included in line 111 ff.

I hope the changes could clarify the misleading parts.

Kind regards.